# The Impact of a CMV Infection on the Expression of Selected Immunological Parameters in Liver Tissue in Children with Biliary Atresia

**DOI:** 10.3390/jcm11247269

**Published:** 2022-12-07

**Authors:** Maria Janowska, Joanna B. Bierła, Magdalena Kaleta, Aldona Wierzbicka-Rucińska, Piotr Czubkowski, Ewelina Kanarek, Bożena Cukrowska, Joanna Pawłowska, Joanna Cielecka-Kuszyk

**Affiliations:** 1Department of Pediatric Surgery and Organ Transplantation, The Children’s Memorial Health Institute, 04-730 Warsaw, Poland; 2Department of Pathomorphology, The Children’s Memorial Health Institute, 04-730 Warsaw, Poland; 3Teva Pharmaceuticals, 00-113 Warsaw, Poland; 4Department of Biochemistry, Radioimmunology and Experimental Medicine, The Children’s Memorial Health Institute, 04-730 Warsaw, Poland; 5Department of Gastroenterology, Hepatology, Nutritional Disorders and Pediatrics, The Children’s Memorial Health Institute, 04-730 Warsaw, Poland; 6Histocompatibility Laboratory, The Children’s Memorial Health Institute, 04-730 Warsaw, Poland

**Keywords:** biliary atresia, cytomegalovirus, cytotoxic T cells

## Abstract

The pathogenesis of biliary atresia (BA) is still not clear. The aim of this study was to evaluate the expression of selected immunological parameters in liver tissue in BA children based on CMV/EBV infection status. Eight of thirty-one children with newly diagnosed BA were included in this prospective study and assigned to two groups (I with active infection, II without active or past infection). All studies were performed on surgical liver biopsies. To visualize CD8+ T cells and CD56 expression, immunohistochemical staining was performed. The viral genetic material in the studied groups was not found, but CMV infection significantly affected the number of CD8+ lymphocytes in both the portal area and the bile ducts. The average number of CD8+ cells per mm^2^ of portal area in Groups I and II was 335 and 200 (*p* = 0.002). The average number of these cellsthat infiltrated the epithelium of the bile duct per mm^2^ in Group I and II was 0.73 and 0.37 (*p* = 0.0003), respectively. Expression of CD56 in the bile ducts corresponded to the intensity of the inflammatory infiltrate of CD8+ cells. Our results suggest that active CMV infection induces an increased infiltration of CD8+ lymphocytes, which could play a role in BA immunopathogenesis. Increased CD56 expression can be a sign of a newly formed bile structure often without lumen, suggesting inhibition of the maturation process in BA.

## 1. Introduction

Biliary atresia (BA) is a progressive cholangiopathy of unclear etiology affecting extra- and intrahepatic bile ducts. It is the most common cause of neonatal cholestasis and is the main indication for liver transplantation in children. The nature of the disease is the obstruction of the biliary outflow from the liver due to progressive inflammation, fibrosis, and proliferation of the intrahepatic bile ducts [1]. Despite many years of research, the etiopathogenesis of the disease is still not fully understood. It is suggested that BA is not a single disease, but rather a phenotypic expression of various specific entities developing as a result of a combination of external (e.g., viruses, toxins), immunological and genetic factors [2,3].

Reports on the temporal–regional concentration of BA cases may support the theory of damage to the bile ducts due to the action of a viral factor in the prenatal period. Of the large list of viruses studied in the pediatric population with BA, cytomegatovirus (CMV) seems to be the most likely causative factor [4]. Fisher et al. demonstrated a higher prevalence of anti-CMV antibodies in mothers of children with BA, a higher concentration of CMV-IgM in infants, and the presence of viral DNA in the liver in half of the studied patients with this disease [5]. Xu Y et al. detected CMV DNA in 60% of a large cohort of Chinese patients [6]. In turn, Brindley et al., in 56% of BA patients (out of a group of 16 patients), observed a significant increase in liver T cells producing interferon-gamma in response to CMV, compared to the control groups. This suggests a previous CMV infection [7]. Davenport et al. found a correlation between CMV infection at diagnosis and the presence of higher parameters of cholestasis, hepatitis and fibrosis, and the need for liver transplantation in a group of 210 CMV IgM (+) children with BA [8]. In 2009, the Hannover Group published the results of a biopsy study of 74 BA patients. Research on the amount of RNA/DNA of hepatotropic viruses showed their presence in nearly 50% of patients (reovirus—33%, CMV—11%, adenovirus—1%, enterovirus—1.5%). It has been suggested that viral infection can play a role in the activation of immune deregulation and loss of tolerance to bile epithelial antigens. The question remains whether viruses are an integral part of the destructive inflammatory process of the biliary tract or if they are of minor importance [9]. The aim of the present study was to evaluate the expression of selected immunological parameters in liver tissue in BA children with active CMV/EBV (Epstein–Barr virus) infection and in children without such an infection.

## 2. Materials and Methods

### 2.1. Patients

After receiving approval from the local ethics committee, we prospectively recruited children with BA who underwent Kasai portoenterostomy between 2014 and 2019. All the patients had complete obliteration of the bile ducts, and none of them presented biliary atresia splenic malformation (BASM) syndrome. In all cases, the surgical liver biopsies were performed during portoenterostomy to obtain tissue for analysis. Age of HPE was presented in weeks and corrected in premature patients.

Liver biochemistry (total and direct bilirubin, aspartate aminotransferase (AST), alanine aminotransferase (ALT), γ-glutamyltranspeptidase (GGT)) and coagulation features (INR) at the time of diagnosis were available from the prospectively maintained database.

All 31 patients were tested for CMV/EBV infection (serum-specific antibodies of immunoglobulins (Ig) M and IgG, and RT-PCR). None of the patients had active EBV infection. Patients with active CMV infection (4 out of 31 children with positive IgM antibodies or/and positive RT-PCR) were selected for Group I. Patients without an active history of CMV/EBV infection (4 out of 31 children with negative IgM and IgG antibodies and RT-PCR) were selected for Group II. Patients’ different virological status (6/31: CMV IgM-IgG+ EBV IgM-IgG-; 9/31: CMV IgM-IgG+ EBV IgM-IgG+; 8/31: CMV IgM-,IgG- EBV IgM-IgG+) did not qualify the subject for any of the above groups.

### 2.2. Determination of EBV and CMV Infection Status

The virological status of patients was based on serological and molecular tests performed prior to hepatoportoenterostomy. Serum IgM and IgG CMV antibodies were measured by a microparticle chemiluminescent immunoassay at the same time. A result was considered positive when serum antibody titers exceeded the cutoff of 1.0 for IgM and 6.0 for IgG. To confirm CMV infection, molecular examination by RT-PCR (urine or serum) was performed. Serological tests for specific IgM and IgG antibodies against viral capsid antigens (VCA) of the EBV virus and antibodies against Epstein–Barr nuclear antigens (EBNA) in the class IgG were performed using an enzyme-linked immunosorbent assay. Serum EBV VCA IgM positivity was defined as serum levels above the cutoff value of 1.0 index.

### 2.3. In Situ Hybridization Technique

The 3µm formalin-fixed paraffin-embedded (FFPE) tissue sections after 16 h at 56 °C were dewaxed twice in Xylen, rehydrated through descending ethanol (100%, 90%, 70%) series, and then rinsed in ultrapure water at room temperature. Preparations were digested in a previously prepared 0.3% endogenous peroxidase with proteinase K (QUIAGEN, Germantown, MD, USA) at a concentration level of 0.3 mg/mL. The hybridization was performed with Histosonda EBER (Cenbimo, Lugo, Spain) and Histosonda Cytomegalovirus CMV (Cenbimo, Lugo, Spain) probes for EBV and CMV, respectively. Slides were incubated for 1 h at 62 °C in ThermoBrite (Abott Molecular, Chicago, IL, USA) for both CMV and EBV. After hybridization, the slides were first incubated with monoclonal mouse anti-digoxin antibody (Cenbimo, Lugo, Spain), and then with labeledpolymerhorseradish peroxidase (HRP) anti-mouse immunoglobulins (Dako EnVision+ System–HRP, Santa Clara, CA, USA). As the substrate, HRP3-amino-9-ethylcarbazole containing hydrogen peroxide was used (Dako EnVision+ System–HRP, Santa Clara, CA, USA). The washed slides were stained with Mayer’s hematoxylin(O.KINDLERMikroskopischeGläser EUKITT, Freiburg, Germany) and sealed with coverslip and Dako Faramount Aqueous Mounting Medium (Dako, Santa Clara, CA, USA). As positive controls for EBV and CMV, in situ hybridization sections from CMV- or EBV-infected liver tissue were used. The negative controls were performed without the specific hybridization probe.

### 2.4. Immunohistochemistry

The immunohistochemistry (IHC) staining reaction was performed on FFPE tissue samples. IHC was performed on the Ventana BenchMark ULTRA IHC/ISH autostaining system using primary antibodies anti-CD8 (Ventana anti-CD8, clone SP57, rabbit monoclonal primary antibody) and anti-CD56 (Cell Marque, clone MRQ-42, rabbit monoclonal antibody) after antigen retrieval in Cell Conditioning 1 buffer followed by detection with the Ultra View HRP system (Roche/Ventana).

### 2.5. Morphometric Analyses

IHC slides were scanned by an Hamamatsu NanoZoomer 2.0 RS scanner (Hamamatsu Photonics, Hamamatsu, Japan) at a magnification of 40× and 20× for morphometric analyses of CD8 and CD56 expression, respectively. Next, morphometric analyses were performed with the use of a Cell^P program (Olympus). To analyze CD8+ cells, the three portal areas were selected for each slide. The surface of the total portal areas and all bile ducts in every portal area were measured. The number of CD8+ cells was counted and calculated per 1 µm^2^ of both the portal area and the bile duct area. Results were presented as an arithmetical mean ± standard deviation (SD).

To analyze anti-CD56 immunostaining, five regions within the portal area with positive CD56 expression were selected for each slide. The results were presented as a percentage of the area with CD56 expression in selected regions in relation to the entire area of the scanned image under 20× magnification (this area was 147,978.43 µm^2^). The method of analysis is illustrated in Figure 1. The threshold parameters were set using the HSI color space model in the following way: hue (H) 90°, saturation (S) 256°, intensity value (I) 123°.

### 2.6. Statistical Analyses

The data were analyzed using the Stata Program version 12.1. The Student’s *t* test and the Mann–Whitney U test were performed to compare the two sets of data depending on the group size and the type of distribution. The Shapiro–Wilk test was used to check if the distribution was normal. A *p* value < 0.05 was considered statistically significant.

### 2.7. Ethics

The approval of the Bioethics Committee at The Children’s Memorial Health Institute was obtained (number of ethical approval 17/KBE/2017). Written informed consent was obtained from the parents or the legal guardians.

## 3. Results

### 3.1. Patients’ Characteristics

Thirty-one Caucasian children (fourteen female, seventeen male) aged 13.1 weeks (range 3,4–31 weeks) with newly diagnosed BA were included in the study. The patients’ general biochemical and anthropological characteristics are presented in Table 1.

Of all these patients, eight were selected for further study and divided into two groups: Group I (*n* = 4, 4 male, 0 female; aged 16 weeks [range 10–28 weeks]), with an active CMV infection, and Group II (*n* = 4, 4 female, 0 male; aged 21 weeks [range 12–31 weeks]), without a past and/or active CMV/EBV infection (Table 2).

Six of eight selected patients underwent LTx (liver transplantation) (three patients in Group I and three patients in Group II). The mean age of LTx patients in Group I was 32 weeks (20 weeks from HPE); in Group II it was 83 weeks (71 weeks from HPE). In Group II, one patient was disqualified from LTx due to neurological complications of prematurity. The patient died in the second year of its life. In Group I, one patient was listed for LTx at the age of 5. Table 2 shows biochemical, anthropological and clinical characteristics of the patients from Group I and Group II.

CMV and EBV infection status was confirmed by the assessment of specific IgM and IgG concentrations and the RT-PCR method in all the children (Table 3). The in situ hybridization technique did not show genetic material of either EBV or CMV in the liver tissues of the study patients. Figure 2 shows a negative in situ hybridization result for the CMV of a patient in Group I.

### 3.2. Histopathology

Histopathology of the liver showed fibrosis, bile clusters in the bile ducts, ductular proliferation and mild/moderate inflammation in all cases (Figure 3a). The presence of periductal and ductal inflammation was documented by H&E staining within the portal tracts: inflammatory infiltrates consisted of granulocytes and lymphocytes found in the wall of bile ducts/ductules and were also distributed in the fibrotic tissue between portal tracts. Bile plugs were also seen in every case and were accompanied by stromal oedema (Figure 3b). Immunohistochemical staining for cytokeratins CK7 (Figure 4a) and CK19 (Figure 4b) was performed in every case.

### 3.3. CD8 Expression

Inflammatory cells were present in the portal tracts lying in the fibrotic tissue and infiltrating the bile ducts and ductules. The infiltration of lymphocytes CD8+ was more prominent in Group I with an active CMV infection compared to Group II without such infection (Figure 5). CD8+ cell infiltration dominated in portal areas in both groups. In the ductules, CD8+ lymphocytes were located between cholangiocytes. Morphometric analyses confirmed a statistically significant increase in the number of CD8+ lymphocytes both in the portal areas (*p* = 0.002) and in the bile ducts (*p* = 0.002) in patients with active CMV infection in comparison with the group of patients without infection (Group II). The number of lymphocytes expressing CD8 separately in the portal tracts and in the bile ducts is presented in Table 4.

### 3.4. CD56 Expression

CD56 as a marker of immature bile ducts was expressed on the biliary epithelium of the bile ducts and bizarre forms of DPM in all cases, but was more prominent in Group I with CMV infection compared to Group II (Figure 6).

The morphometric analysis confirmed statistical significance in CD56 expression between the groups (*p* = 0.00003) (Table 5).

## 4. Discussion

Abnormal immune response in the pathogenesis of BA was reported before with the special role of CMV infection as an external trigger [1,10,11,12].

In our study, we presented a significant increase in CD8+ lymphocytes in both portal areas and bile ducts in patients with an active CMV infection vs. CMV-negative patients. This is in line with previous reports on the cytotoxicity of lymphocytes towards the biliary tract [11,13,14,15]. Mack et al. described the results of the transplantation of CD3+ cells (probably containing CD4+, CD8+, NK cells) into adult SCID mice. The transplanted cells lodged in the bile ducts, causing inflammation without losing the lumen of the ducts [11]. Shivakumar et al. showed that in response to a viral agent (Rotavirus Rhesus), CD8+ cells damage the bile ducts, leading to a BA-like phenotype in newborns. Interestingly, in CD8+-deficient mice, the disease did not progress and the bile duct lumen remained continuous [15].

Another study showed that the lymphocytic infiltration of the bile ducts in BA patients consists mainly of CD8+ T cells and NK cells. CD8+ cells can damage the bile duct epithelium as a result of NK cell activity [13]. The involvement of NK cells in biliary diseases has been repeatedly described [13,16,17,18]. CD56 glycoprotein is the differentiation antigen for these cells. One of the tasks of NK cells is to respond to infectious provocations in the liver. The proliferation and activation of these lymphocytes lead to the destruction of virus-infected cells [19]. In the study by Guo et al., abundant NK lymphocyte infiltration was found in the extrahepatic bile ducts of neonates with BA, including patients with CMV infection, compared to the controls [13]. We observed a similar relationship in our study. There was higher expression of CD56 in BA patients with a CMV infection than in those without infection.

On the other hand, CD56 is also a stem cell marker associated with biliary differentiation and ductal reaction [20], and it is considered an additional marker in the diagnosis of BA [21,22,23,24,25]. However, the increased expression of CD56 has also been observed in other cholestatic diseases, such as choledochal cyst and progressive familial intrahepatic cholestasis [25]. Zhang et al. observed in their study a positive correlation of CD56 expression with liver fibrosis [23]. Similar conclusions were presented by Ayyanara et al. [25]. We suggest that an increased expression of CD56 found in the portal space (including bile ducts in children with active CMV infection compared to children without infection) can be a sign of a newly formed bile structure often without lumen, suggesting the inhibition of the maturation process in BA. It is probable that CMV infection in patients with BA worsens the prognosis of the disease. In our study, patients with CMV infection required LTx at a much younger age than those in the group without CMV infection (32 vs. 83 weeks).

### Study Limitations

Our study is limited by the small size of our group of patients. However, it is necessary to emphasize that three out of eight patients were extremely premature infants. Due to extreme prematurity and its complications, the diagnosis of BA in this group is difficult. For this reason, and due to the late referral of the patient to the reference center, KPE is performed in these patients at a later age compared to full-term newborns [26]. Other limitations include the lack of a control group and patients with other cholestatic diseases, non-BA and no confirmation of an increase in CD8+ number in peripheral blood, as well as no double immunohistochemical staining to visualize CD8+ lymphocytes in liver tissue co-stained with CD56, and cytokeratins such as CK7 or CK19. Immune profiles are different in the early and late stages of response [27], but in our study we focused on late-stage immune cells (CD8+ and CD56+), which is another limitation of our study. Further studies on a larger group of patients are needed to assess the role of individual effector cells in the pathogenesis of BA and the role of CMV infection in this process.

## 5. Conclusions

In conclusion, our observations indicate that active CMV infection induces an increased infiltration of cytotoxic CD8+ cells that could play role in BA immunopathogenesis. The expression of CD56 is not usually present in the mature biliary epithelium, but appears in the reactive and proliferative biliary epithelium. CD56+ can be a sign of a newly formed bile structure often without lumen, suggesting the inhibition of the maturation process in BA.

## Figures and Tables

**Figure 1 jcm-11-07269-f001:**
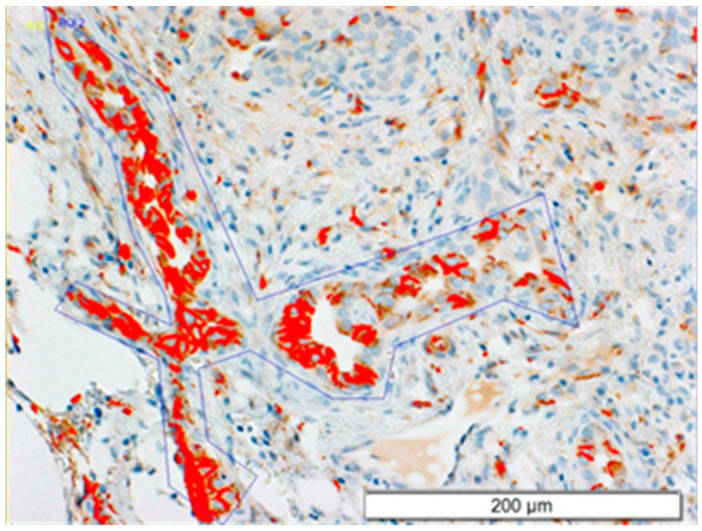
An example of estimation of the expression of CD56 in the portal area of the chosen regions (20× magnification, 147,978.43 µm^2^).

**Figure 2 jcm-11-07269-f002:**
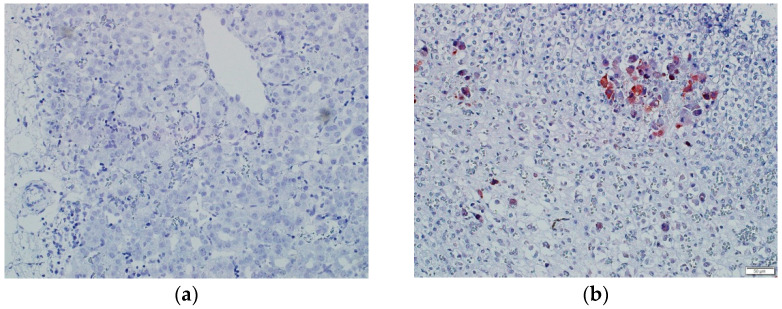
In situ hybridization for CMV in the liver tissue. The patient from Group Ishowing a negative reaction (**a**) and the positive control for CMV (**b**).

**Figure 3 jcm-11-07269-f003:**
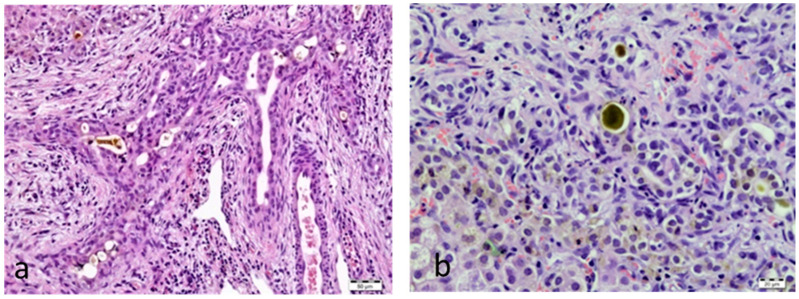
Bile duct and ductular proliferation in the portal tract (**a**). Lymphocytic and granulocytic infiltrates in the portal tract, bile clusters in the bile ducts, and fibrosis (Ishak fibrosis score 4) in the patient from Group I (**b**). Hematoxylin and eosin: 250× (**a**), 500× (**b**).

**Figure 4 jcm-11-07269-f004:**
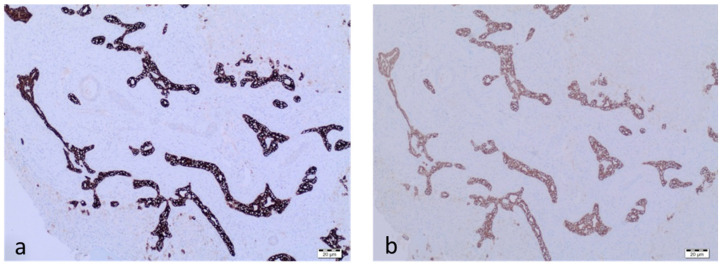
Immunohistochemical staining for CK7 (**a**) and CK19 (**b**) fromthe patient from group I.

**Figure 5 jcm-11-07269-f005:**
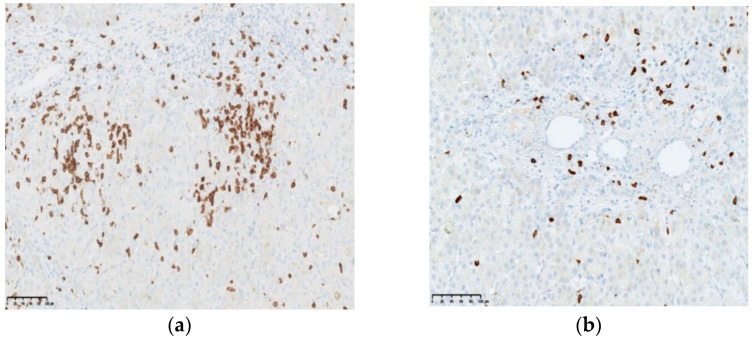
Presentation of the inflammatory infitrates of CD8+ cells in the portal spaces, including bile ducts. IHC (Ventana, anti-CD8, clone SP57). Liver tissue from a patient with BA and with active CMV infection (**a**) and without active or past CMV/EBV infection (**b**).

**Figure 6 jcm-11-07269-f006:**
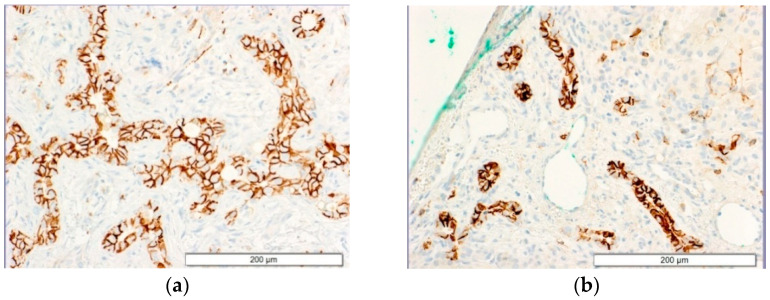
Expression of CD56 in the portal spaces. IHC (Cell Marque, anti-CD56, clone MRQ-42). Liver tissue from a patient with BA and with an active CMV infection (**a**) and without an active or past CMV/EBV infection (**b**).

**Table 1 jcm-11-07269-t001:** The patients’ general biochemical and anthropological characteristics before the HPE.

	Hbd (Weeks)	Age of HPE (Weeks) *	Total Bilirubin (mg/dL)	Direct Bilirubin (mg/dL)	ALT (U/L)	AST (U/L)	GGTP (U/L)	INR
Q1	36	8.1	6.44	5.8	82	134	302	1.03
median	39	11.0	8.74	7.22	128	177	457	1.07
Q3	39.5	13.9	10.39	9.11	175	248.5	784	1.1

ALT—alanine aminotransferase; AST—aspartate aminotransferase; GGTP—γ-glutamyl transpeptidase; INR—International Normalized Ratio; Q—quartile; Hbd—hebdomas—week of pregnancy; HPE—hepatoportoenterostomy. * Age corrected in premature patients.

**Table 2 jcm-11-07269-t002:** Biochemical, anthropological and clinical characteristics of the patients from Group I and Group II before the HPE.

	Patient	Sex	Hbd (Weeks)	Age of HPE (Weeks) *	Total Bilirubin (mg/dL)	Direct Bilirubin (mg/dL)	ALT (U/L)	AST (U/L)	GGTP (U/L)	INR	Age of LTx (Weeks) *	Time between LTx and HPE (Weeks)
Group I	1	M	40	12	6.31	7.01	88	143	820	1	29	17
	2	M	40	13	8.34	7.47	128	202	1616	1.05		
	3	M	40	10	9.37	7.92	225	358	314	1.08	38	28
	4	M	27	15	11.66	10.24	271	439	450	1.28	28	13
Group II	5	F	38	12	9.46	8.27	176	228	1652	1.08	43	29
	6	F	39	11	5.91	5.24	104	177	358	1.07	41	30
	7	F	26	11	10.5	9.3	544	685	204	1.15	166	155
	8	F	24	15	14.44	12.62	271	326	754	1.07	dq	

ALT—alanine aminotransferase; AST—aspartate aminotransferase; dq—disqualified from LTx; GGTP—γ-glutamyl transpeptidase; Hbd—hebdomas—week of pregnancy; HPE—hepatoportoenterostomy; INR—International Normalized Ratio; LTx liver—transplantation. * Age corrected in premature patients.

**Table 3 jcm-11-07269-t003:** Virological status of CMV infection in the study groups: Group I with active CMV infection and Group II without active CMV infection.

	Patient	CMV		
		IgM (AU/mL)	IgG (AU/mL)	PCR
Group I	1	0.57	59.7	(+)
	2	7.5	41.5	(+)
	3	0.79	715.2	(+)
	4	8.9	1181	(+)
Group II	5	0.23	3.0	(−)
	6	0.13	1.5	(−)
	7	0.15	0.53	(−)
	8	0.25	0.4	(−)

CMV IgM: positive >= 1.0; negative < 1.0. CMV IgG: positive >= 6.0; negative < 6.0.

**Table 4 jcm-11-07269-t004:** The number of CD8+ cells in the liver tissue of children with active CMV infection (Group I) and without CMV/EBV infection (Group II).

	The Surface of Slides (µm^2^)	The Surface of Portal Areas (µm^2^)	The Number of CD8+ Cells Per µm^2^ of Portal Areas	The Surface of Bile Ducts in Portal Areas (µm^2^)	The Number of CD8+ Cells Per µm^2^ of Bile Ducts
Group I	17.67 ± 7.86	0.62 ± 0.35	206.92 ± 82.01 * *p =* 0.0019	2870.55 ± 3279.32	0.73 ± 1.23 # *p =* 0.0019
Group II	17.99 ± 5.06	0.41 ± 0.28	82.00 ± 38.98	2089.63 ± 1973.30	0.37 ± 0.62

The results are presented as arithmetical means ± SD. * Statistically significant difference in portal areas between the studied groups analyzed with the use of Student’s *t* test, *p* = 0.0019 (d.f. = 18); **#** Statistically significant difference in bile ducts between studied groups analyzed with the use of the Mann–Whitney U test, *p* = 0.0019.

**Table 5 jcm-11-07269-t005:** The CD56 expression in the livers of children with an active CMV infection (Group I) and without an active CMV/EBV infection (Group II).

Group I (*n* =4) (%)	Group II (*n* =4) (%)	*p*-Value
2.92 ± 1.43	1.43 ± 0.28	0.00003

The medium percentage (%) ± standard deviation of CD56 expression was measured as described in Material and Methods. Statistical differences were calculated with the use of the Mann–Whitney U test.

## Data Availability

Not applicable.

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
