# Peer review of "The Impact of a CMV Infection on the Expression of Selected Immunological Parameters in Liver Tissue in Children with Biliary Atresia"

_jcm, 2022, doi:10.3390/jcm11247269_

Round 1
Reviewer 1 Report (Previous Reviewer 3)
Authors answered all my questions. The paper could be accepted.
Author Response
Authors would like to thank you for review of our manuscript.
Reviewer 2 Report (New Reviewer)
The authors reported the expression of selected immunological parameters in liver tissue in BA children with active CMV / EBV infection.
They claimed that active CMV infection induces, increased infiltration of CTLs that could play a role in BA immunopathogenesis. Additionally, they concluded CD56+ can be a sign of a newly formed bile structure often without lumen, suggesting inhibition of the maturation process in BA.
Overall, I believe that this manuscript can be important because it is known that a correlation between CMV infection at diagnosis and the prognosis in the group of CMV IgM (+) patients with BA. This aspect is a minor condition and there are a few pieces of evidence. Thus, there is a lot of problems in this area.
2.3. In situ hybridization technique
They indicated in situ hybridization technique in methods. They should indicate the data, even if it is Supplementary. It is important to show the localization of CMV in liver tissue, taking into account that this study is based on the assumption of CMV infection.
3.1. Patients’ characteristics
Tables 1 and 2
In abbreviations, Hbd and HPE should be indicated.
Figure1 in page7
It should be changed in Figure 2.
3.2.CD8 expression
I understand an increase in the number of CD8+ lymphocytes in patients with active CMV infection in comparison with the group of patients without infection with morphometric analyses. I recommend the authors to add more data on CD8 for establishing evidence. How about adding a CD8 blood test and/or qPCR data on liver tissue?
If possible, could you co-stain with bile duct marker, CK7 or CK19, etc.? I recommend indicating the localization of cholangiocytes and CD8+ lymphocytes.
3.3. CD56 expression
I agree that CD56 was more prominent in Group I with CMV infection compared to Group II. I expect this finding indicates the correlation between the severity of bile duct proliferation and the prognosis of BA patients with CMV. I believe the prognosis of BA patients with CMV is worse than patients without CMV. Could the authors add the clinical data of both Group1 and 2? I understand bile duct proliferation correlates with the severity of BA. For indicating the feasibility of CD56, authors should indicate their clinical data including prognosis.
I expect that immature bile ducts were detected in Figure 4. However, could the authors stain with CK7 or CK19 also? The findings is not understood clearly.
4. Discussion
I think it would be easier to understand if the authors describe the discussion in separate paragraphs.
Author Response
Authors would like to thank you for review of our manuscript and all comments and suggestions which allow us to improve the paper. All changes made in the manuscript are marked in yellow.
Please, find attached our answers and edited manuscript.

This manuscript is a resubmission of an earlier submission. The following is a list of the peer review reports and author responses from that submission.
Round 1
Reviewer 1 Report
This study investigated the role of CMV/EBV (Ebstein-Barr virus) in the Biliary atresia by immunological profiling.
Main:
1. Please put the Figure1 in the result sections (main or supplementary figure).
2. Please put the Figure2 in the result sections (main or supplementary figure).
3. Table1 only presents 8 patients, how about the remaining ones?
4. Table2 only presents 8 patients, how about the remaining ones?
5. Is there any correlation between the CMV infection and risk of the Biliary atresia?
6. Is there any correlation between the CMV infection and fibrosis stage?
7. Is there any correlation between the CMV infection and the liver/bile duct damage?
8. Based on the study (Yang C, et al. Front. Pediatr. 10:902571), the immune profiles are different between early and late stage. For example, activated mast cells, eosinophils, and neutrophils were more abundant in inflamed livers than in fibrotic livers. In contrast, CD8+ T cells and γδT cells were more abundant in fibrotic livers. Your study mainly focused on the late stage immune cells. How about the early stage profiling?
Reviewer 2 Report
This is an excellent study; please add a degree of freedom for each p-value. Also, add a scale bar, annotations, and type of dye or stain for each figure. It is too early to conclude this objective, so double-blind randomized studies are needed to prove these conclusions.
Reviewer 3 Report
Authors reported 31 BA in whom immunohistochemical staining was performed only in 8 (4 with CMV and 4 without CMV infection). This in the main and inherent limitation of the study; the proper size of the population should be stated in the abstract, that is misleading. Other suggestions:
- please insert number of ethical committee;
- how did you select the 8 pmts from the population of 31?
- According to your caption, Table 1 should be referred to all 31 patients, and not to the subgroup of 8;
- quite surprising, the age at KPE is rather high: it seems that no patient undergone surgery between 12 weeks, and some at 25-28-31 weeks. Do you perform KPE on 217-day-old BA patient? How do you explain that?
- In patient with CMV-related BA, cirrhosis and endothelial damage is somewhat time-dependent. How could you compare immunoistochemical expression in patient ranging from 12 to 20 weeks? Moreover, the control group in sensibly older (5 weeks).
- methodology of the study is well-done and clearly presented. My suggestion is to increase the sample size and apply this nice methodology to a proper group of patients.